# Sleep Duration, Midday Napping, and Serum Homocysteine Levels: A Gene–Environment Interaction Study

**DOI:** 10.3390/nu15010210

**Published:** 2023-01-01

**Authors:** Tingting Mo, Yufei Wang, Hui Gao, Wending Li, Lue Zhou, Yu Yuan, Xiaomin Zhang, Meian He, Huan Guo, Pinpin Long, Tangchun Wu

**Affiliations:** Department of Occupational and Environmental Health, Key Laboratory of Environment and Health, Ministry of Education and State Key Laboratory of Environmental Health (Incubating), School of Public Health, Tongji Medical College, Huazhong University of Science and Technology, Wuhan 430030, China

**Keywords:** sleep duration, midday napping, homocysteine, one-carbon metabolism, gene–environment interaction, genetic susceptibility

## Abstract

The associations of sleep duration and midday napping with homocysteine (Hcy) levels, and whether these sleep behaviors modify the association between genetic predisposition and Hcy levels, has yet to be investigated. We included 19,426 participants without severe health conditions at baseline from the Dongfeng–Tongji cohort. In a subgroup of 15,126 participants with genetic data, a genetic risk score (GRS) based on 18 Hcy-related loci was constructed to test the gene–sleep interactions in Hcy. Hcy levels were higher in subjects with a long sleep duration (≥9 h) and midday napping (>90 min), as compared to those who reported a moderate sleep duration (7 to <8 h) and midday napping (1–30 min) (all *p* values < 0.05). A long sleep duration and midday napping showed a joint effect in increasing Hcy (*p* for trend < 0.001). Significant interactions regarding Hcy levels were observed for a long sleep duration with GRS and MTHFR rs1801133, and long midday napping with DPEP1 rs12921383 (all *p* values for interaction < 0.05). Overall findings indicated that a long sleep duration and midday napping were associated with elevated serum Hcy levels, independently and jointly, and amplified the genetic susceptibility to higher Hcy.

## 1. Introduction

One-carbon metabolism (OCM) consists of various biochemical reactions that transfer one-carbon units between different sites [1], and involves multifarious physiological processes, including amino acid homeostasis, epigenetic maintenance, nucleic acid synthesis, and redox regulation [2]. During these processes, OCM donates a methyl group for homocysteine (Hcy) remethylation to methionine [3]. Hcy is a downstream product of OCM and can flexibly reflect disorders in OCM [4,5]. Elevated Hcy levels are associated with various diseases [6,7], especially cardiovascular diseases (CVDs), with the heaviest burden on global public health [8]. The investigation of modifiable behavioral factors regarding Hcy could provide a crucial insight into mechanisms and diseases regarding Hcy and OCM, as well as their related diseases.

Sleep, a typically indispensable behavioral factor in a normal daily routine, is essential for human health. Surveys of US adults reported significant inverse associations of serum vitamin B12 (indispensable cofactor of OCM) with sleep duration, and folate with sleep distribution [9], and found a significant relation between extremely short sleep durations (<5 h) and elevated Hcy levels [9,10]. However, both studies did not adjust for midday napping, and the association between midday napping and Hcy levels has never been explored. Midday napping, commonly practiced in countries and areas with warmer climates [11,12,13], is especially prevalent among the elderly population [14]. The relations of sleep duration and midday napping separately and jointly with Hcy, as well as the potential mechanisms, have yet to be elucidated.

In recent decades, gene–environment interaction has become an effective tool in assessing hypothesized biological mechanisms [15,16]. The hitherto largest meta-analysis of Hcy-related genome-wide association studies (GWAS) reported 18 Hcy-related single-nucleotide polymorphisms (SNPs) (N = 44,147) [17], which might play important roles in Hcy regulation. However, no study has investigated the interactions of Hcy-related genetic variations with sleep duration and midday napping on Hcy levels, and the underlying mechanisms remain largely unknown.

Thus, on the basis of the Dongfeng–Tongji (DFTJ) cohort, we investigated the relations of sleep duration and midday napping with Hcy levels. Furthermore, we constructed the genetic risk score (GRS) based on 18 well-established Hcy-related loci [17], and tested the interactions of sleep duration and midday napping with genetic susceptibility on Hcy levels.

## 2. Methods

### 2.1. Participant Inclusion

The DFTJ cohort, established on 1 September 2008, with ongoing follow-ups (every 5 years), is a population-based prospective survey of retirees from the Dongfeng Motor Corporation (DMC) [18]. The current study included 38,295 subjects from the first follow-up in 2013, when baseline information was collected and serum Hcy levels were measured. We included 28,041 subjects without coronary heart disease, stroke, cancer, and severe abnormalities in electrocardiogram, and then excluded 8615 subjects with missing data on Hcy and sleep behaviors, leaving 19,426 participants for principal analyses. For genetic analyses, we restricted the population to a subgroup of 15,126 subjects with genetic data. The DFTJ cohort has approval from the Ethics Committee of the School of Public Health, Tongji Medical College, Huazhong University of Science and Technology (2012-10). Written informed consent was provided by each subject.

### 2.2. Ascertainment of Sleep

In the DFTJ, information on sleep was collected by self-administrated questionnaires. For sleep duration, participants were asked, “In the past six months, when did you normally go to bed at night and wake up in the morning?” To assess midday napping, participants were asked, “Were you in the habit of having midday napping in the past six months?” For those who answered “Yes”, the duration of napping was obtained. Sleep duration was derived based on bedtime and wake time. To be consistent with our previous studies with respect to sleep [19,20], and to ensure sufficient statistical power with a sufficient sample size in each category, sleep duration was divided into <7, 7 to <8, 8 to <9, or ≥9 h, while midday napping was categorized as none, 1–30, 31–60, 61–90, or >90 min.

### 2.3. Measurement of Serum Homocysteine Levels

After overnight fasting, blood samples were collected in ethylene diamine tetra-acetic acid (EDTA) tubes, centrifuged, and stored at −80 °C for subsequent use. We used the Abbott Architect i2000 Automatic Analyzer (Abbott Park, IL, USA) with standard hospital assays of chemiluminescence reaction to examine serum Hcy levels. The coefficient of variation was tested (intra-assay = 5.8% and inter-assay = 6.5%).

### 2.4. Covariates

Detailed socio-demographics, lifestyle data, and personal medical histories were obtained from semi-structured questionnaires [18]. Participants also completed various physical and biological measurements at baseline according to a standard protocol [18].

Covariates of this study included age, sex (female, male), body mass index (BMI), estimated glomerular filtration rate (eGFR), education levels (primary school or below, middle school, high school or beyond), presence of hypertension, dyslipidemia, and diabetes (yes, no), smoking status (current, ever, never), drinking status (current, ever, never), dietary intake (meat, milk or dairy products, beans or soy products, fish or seafood, and fruits or vegetables; ≥5 times/week; yes, no), regular exercise (yes, no), snoring (yes, no), and sleep quality (good, fair, poor). Definitions of covariates are detailed in the Appendix A.

### 2.5. Genotyping and Genetic Risk Score

We selected 18 independent SNPs on the basis of the largest GWAS meta-analysis on Hcy (N = 44,147) [17]. The genotyping was performed using Illumina Infinium OmniZhongHua-8 Chips, and the processes of genotyping and quality control have been detailed previously [21]. Each SNP passed the Hardy–Weinberg equilibrium test (*p* > 0.05) for the study population. All SNPs were re-coded into 0, 1, or 2 by the amount of Hcy-increasing alleles. Weighted GRS was calculated based on β coefficients obtained from the reported meta-analysis [17], using the equation
(1)$$GRS=\left(\sum_{i=1}^{n=18}\beta_{i}SNP_{i}\times 18\right)÷\sum_{i=1}^{n=18}\beta_{i}$$

As sensitivity analyses, the GRS-6 was constructed using 6 SNPs verified within this study (*p* < 0.05/18), with the β coefficients obtained from our population.

### 2.6. Statistical Analyses

To approximate normal distributions, serum Hcy levels were subjected to natural logarithmic transformation (ln-transformed). We assessed the relations of sleep duration and midday napping with Hcy by generalized linear regression models, with the reference groups set as 7 to <8 h of sleep duration and 1–30 min of midday napping, respectively, as previous studies suggested that these were beneficial [22,23]. We used three set of adjustments for potential confounders: model 1, adjustments for age and sex; model 2, further adjustments for BMI, eGFR, education levels, hypertension, dyslipidemia, diabetes, smoking status, drinking status, dietary intake, regular exercise, snoring, and sleep quality; model 3, additional adjustment for alternative sleep behaviors, namely midday napping or sleep duration, accordingly.

Stratified analyses based on model 3 were performed by age (<65, ≥65 years), sex (female, male), BMI (<24, ≥24 kg/m^2^), eGFR (<60, ≥60 mL/min/1.73 m^2^), never smoking (yes, no), never drinking (yes, no), and the presence of hypertension, dyslipidemia, and diabetes (yes, no). Specifically, to estimate the potential joint effect of sleep duration as well as midday napping in increasing Hcy levels, we calculated the multivariable-adjusted mean Hcy values via the combined categories of sleep duration and midday napping.

We also performed sensitivity analyses by additionally adjusting for job category (manufacturing or manual labor work, services or sales work, office work) and duration of past shift work (none, ≤5.00, 5.25–10.00, 10.50–20.00, and >20.00 years) based on model 3. In the extended analysis, we also combined the sleep duration and midday napping into a 24-h sleep duration, which was further divided into 4 categories, i.e., <8, 8 to <9, 9 to <10, or ≥10 h, taking 8 to <9 h as the reference group. The β (95% CI) of serum Hcy levels was calculated using generalized linear models with adjustments for covariates used in model 1, model 2, and the sensitivity analysis.

To test the effect of each SNP on Hcy, we performed a principal component analysis for the correction of population stratification using PLINK 1.9 [24], and we adjusted for age, sex, and the first 10 principal components in the linear regression models. Generalized linear models were used to estimate the difference in Hcy levels by GRS (tertiles and per-5 risk alleles) with adjustments for age and sex.

We further performed gene–sleep interaction (GRS and SNPs) analyses regarding Hcy levels. The GRS was divided into three subgroups by tertiles for the study population. The multivariable-adjusted mean Hcy values were calculated by the combined categories of GRS and sleep duration, as well as GRS and midday napping. Based on model 3, gene–sleep interactions were tested by modeling product terms of gene (GRS and genotypes) with sleep duration and midday napping, separately. As sensitivity analyses, all genetic analyses were also performed on GRS-6.

Statistical analyses were conducted using SAS version 9.4; Institute, Gary, NC, USA) and R (version 4.1.3; R Core Team). A two-sided *p* < 0.05 was considered statistically significant.

## 3. Results

Among the 19,426 participants, 27.4% reported a long sleep duration (≥9 h) and 5.9% reported long midday napping (>90 min), as shown in Table 1. Appendix A shows the baseline characteristics of the 15,126 participants with genetic data. The GRS ranged from 5 to 25, and the GRS-6 ranged from 0 to 12. No substantial difference was found between the total population and those with genetic data, except a minor difference in sex (percentage of females: 58.0% vs. 56.9%) and age (mean: 62.9 vs. 63.2) (all *p* values < 0.05; Table 2).

In model 3 (Table 3), compared with sleeping for 7 to <8 h, the β (95% confidence intervals (Cis)) of ln-transformed Hcy were 0.005 (−0.021, 0.031) for <7 h, 0.013 (−0.001, 0.029) for 8 to <9 h, and 0.046 (0.030, 0.062) for ≥9 h, respectively. Compared with napping for 1–30 min, the β (95% CIs) of ln-transformed Hcy were 0.015 (−0.002, 0.032) for no napping, 0.005 (−0.014, 0.024) for 31–60 min, −0.007 (−0.032, 0.017) for 61–90 min, and 0.033 (0.005, 0.061) for >90 min, respectively. Stratified analyses showed that the associations between Hcy and long midday napping (>90 min) were more pronounced in subjects who were male, ever smokers, and ever drinkers (Appendix A). Other stratified analyses yielded no significant result.

In the sensitivity analysis, additional adjustment for occupational factors (including job category and duration of past shift work) did not substantially change the results (all *p* values <0.05; Appendix A). Furthermore, we observed that a 24-h sleep duration ≥10 h was significantly associated with higher serum Hcy levels, as compared with a 24-h sleep duration <8 h (Appendix A). These findings confirmed the robustness of our results regarding the associations of sleep duration and midday napping with serum Hcy levels.

Moreover, we observed a significant joint effect of a long sleep duration and midday napping in increasing Hcy. Among the 16 combined categories of sleep duration and midday napping, Hcy levels were lowest among participants reporting a moderate sleep duration (7 to <8 h) and midday napping (1–30 min), and were highest among those with a long sleep duration (≥9 h) and midday napping (>90 min) (*p* for trend < 0.001; multivariable-adjusted mean [standard error, SE] = 14.98 [0.38] and 16.71 [0.49] μmol/L, respectively; Appendix A).

Appendix A presents the details of the selected SNPs in the present study and the reported meta-analysis. The GRS was positively associated with Hcy levels (all *p* values < 0.001; Appendix A). With adjustments for age and sex, the difference (95% CI) in ln-transformed Hcy per 5-risk allele increment was 0.162 (0.149, 0.175) for GRS. Sensitivity analysis on GRS-6 showed similar results (all *p* values < 0.001; Appendix A). Furthermore, we observed a significant interaction between GRS and a long sleep duration on Hcy (Figure 1). Hcy levels were highest among participants with a long sleep duration (≥9 h) and higher GRS and were lowest among those with a moderate sleep duration and lower GRS (multivariable-adjusted mean [SE] = 13.87 [0.51] and 18.09 [0.51] μmol/L, respectively; *p* for interaction = 0.009). No significant interaction was found between GRS and midday napping on Hcy (*p* for interaction = 0.53). We observed similar trends for the combined categories of GRS-6 with sleep duration and midday napping (Appendix A).

Compared to individuals with rs1801133-CC genotypes, carriers of rs1801133-TT+CT genotypes showed stronger relations between long sleep durations and Hcy; meanwhile, compared to carriers with rs12921383-TT genotypes, the relation between long midday napping and Hcy was more pronounced among those with rs12921383-CC+CT genotypes (all *p* values for interaction < 0.05; Figure 2).

## 4. Discussion

To our knowledge, the current study is the first to investigate the associations of sleep duration, midday napping, and gene–sleep interactions with Hcy levels jointly. We found significant relations of long sleep durations (≥9 h) and midday napping (>90 min) with elevated Hcy levels, separately and jointly. Furthermore, a long sleep duration amplified the effect of genetic susceptibility for elevated Hcy levels. Specifically, rs1801133 on the *methylenetetrahydrofolate reductase* (*MTHFR*) gene interacted with a long sleep duration, while rs12921383 on the *Dipeptidase 1* (*DPEP1*) gene interacted with long midday napping in increasing Hcy levels.

To date, only two studies from the same US survey have investigated the relation regarding sleep duration and Hcy, which consistently showed a significant relation between an extremely short sleep duration (≤5 h) and elevated Hcy levels [9,10]. Both studies included participants with a CVD diagnosis at baseline, recorded sleep durations in integers directly, had relatively small sample sizes for groups reporting extremely short/long sleep durations (≤5/≥9 h: 114/173 and 677/365, respectively), and lacked information on midday napping. By contrast, considering the close relations of sleep and Hcy with CVD [25,26], our study included 19,426 middle-aged and elderly Chinese individuals without severe chronic diseases, including CVD. We used precise bedtimes and wake times to calculate the sleep duration and applied a different category to ensure sufficient statistical power with larger sample sizes (<7/≥9 h: 1132/5330). Importantly, we adjusted midday napping when assessing the relation between sleep duration and Hcy, which reinforced the robustness of our results. Similarly, a study consisting of 116,632 participants from 21 countries (including Asian and Western countries) found that, in fully adjusted models, a longer sleep duration (>8 h) was significantly associated with higher risks of major cardiovascular events and death, and such significance was not observed for short sleep durations (<6 h) [27]. Since Hcy has been considered an independent risk factor for cardiovascular and atherothrombotic diseases [28,29], our findings were in line with results of previous studies regarding the significant relation between a long sleep duration and higher risks of CVDs [19,20].

This is the first study to find a significant relation between long midday napping (>90 min) and elevated Hcy levels, independent of sleep duration. Studies conducted in Western countries usually include midday napping in a 24-h sleep duration [30,31,32], and the effect of midday napping on public health is less studied. However, midday napping is common in countries with temperature rises caused by the afternoon sun, such as Asia and Africa, and is of higher prevalence among elderly population [14]. A survey was conducted in 2015 on 7469 Chinese elderly individuals, of which 59.3% reported habitual midday napping [33]. It has been argued that midday napping and nighttime sleep might mutually change or reflect the change in each other, but a conclusion has not been drawn yet [34,35]. We found that the effect of midday napping and sleep duration was independent and they could be joined together in increasing Hcy levels. Our results were in line with the previous studies of the DFTJ cohort that observed a similar independent effect of midday napping, as well as a joint effect with sleep duration, regarding CVDs [19,20]. Altogether, our findings emphasize that sleep duration and midday napping should be studied separately, as well as jointly, when assessing their associations with Hcy levels, especially in Asian areas where midday napping is habitual [27].

The mechanisms underlying these relations are largely unknown. Based on the results of gene–sleep interaction, we suggest that a long sleep duration and midday napping amplified the genetic susceptibility to increasing Hcy levels through their effects on OCM and subsequently affected DNA methylation. Hcy was mainly metabolized through remethylation to methionine and transsulfuration to cysteine (Appendix A) [36]. During remethylation, under the action of MTHFR and cofactor vitamin B12, Hcy receives a methyl group produced from OCM and is remethylated back to methionine, and methionine could be converted into Hcy with the donation of a methyl group for cellular methylation. Both OCM and DNA methylation may be affected by long sleep durations [37,38]. People with a long sleep duration usually had lower levels of vitamin B12 [37], which could lead to Hcy accumulation and reduced cellular methylation. Furthermore, a longer sleep duration contributed to altered DNA methylation among the elderly [39]. In another aspect, long midday napping may increase Hcy levels with the action of DPEP1. During the transsulfuration pathway, Hcy is converted to cysteine, which could be subsequently transformed into glutathione [40]. The regulation of glutathione involves DPEP1 [41,42], which might interfere with Hcy metabolism with its effects on cysteine. Patients with DPEP1 deficiency reported increased urinary excretion of cysteine [43]. A human trial showed that supplementation of cystine (the oxidized form of cysteine in extracellular space) attenuated the Hcy elevation [44], which could be explained by a feedback mechanism of cysteine stimulating Hcy remethylation [45]. Moreover, both methionine and cystine were found to be correlated with self-reported midday napping among postmenopausal women [46]. Experimental studies are encouraged to elucidate the effects and biological mechanisms of long sleep duration and midday napping on OCM and DNA methylation.

The present study has several strengths. Our study is the first to analyze the associations of sleep duration and midday napping with Hcy jointly. With the detailed information of the DFTJ cohort, we were able to perform comprehensive adjustments for potential confounding factors and stratified analyses. The large sample size provided a good opportunity to explore the gene–sleep interaction. For the first time, we investigated whether a long sleep duration and midday napping modified the genetic susceptibility regarding Hcy levels, and proposed probable mechanisms.

Nevertheless, limitations need to be addressed. First, sleep duration was calculated manually from questionnaires. However, it was not practicable to record biological information on sleep in such a large cohort study, while using self-reported questionnaires is common practice in studies of sleep [32,47,48]. Second, the number of subjects with extremely short sleep durations (≤5 h) in our study population was relatively small (*n* = 87); therefore, the relation between extremely short sleep durations and Hcy levels remains to be investigated. Third, the cross-sectional study design hindered us in concluding on causality. Fourth, the study was conducted among middle-aged and elderly Chinese adults; extrapolation of our findings to different populations should be performed with prudence. Finally, replication studies with more comprehensive information, verifying our results and exploring the underlying biological mechanisms, are needed.

## 5. Conclusions

We found a potential detrimental role of a long sleep duration and midday napping in increasing Hcy, and these effects might enhance the genetic susceptibility to higher Hcy levels. Our findings suggest that the disrupted OCM and DNA methylation may be a potential mechanism in the adverse influence of a long sleep duration and midday napping.

## Figures and Tables

**Figure 1 nutrients-15-00210-f001:**
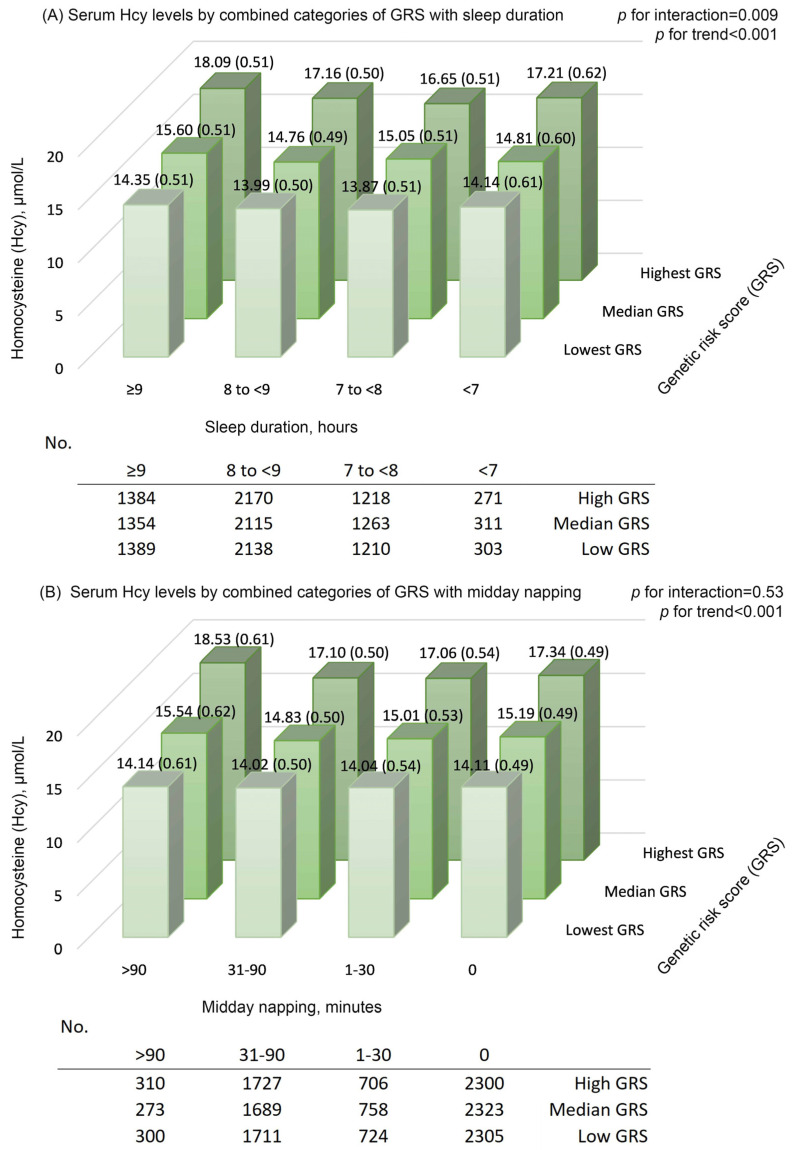
Serum homocysteine (Hcy) levels by combined categories of genetic risk score with (**A**) sleep duration and (**B**) midday napping. Means (standard error) were calculated with adjustments for age (continuous), sex (female, male), education level (primary school or below, middle school, high school or higher), body mass index (continuous), presence of hypertension, dyslipidemia, and diabetes (yes, no), smoking status (current, former, never), drinking status (current, former, never), dietary intake of meat, milk or dairy products, beans or soy products, fish or seafood, and fruits or vegetables (≥5 times/week; yes, no), regular exercise (yes, no), snoring (yes, no), and sleep quality (good, fair, poor); in addition, sleep duration (continuous) was adjusted for investigation of the GRS–napping interaction in Hcy, and midday napping (continuous) was adjusted for investigation of the GRS–sleep interaction in Hcy. The GRS (lowest, median, or highest group) groups were defined by tertiles for the total population. *p* for interactions was calculated by including the product term of GRS with sleep duration or midday napping in the multivariable-adjusted model, separately. GRS, genetic risk score; Hcy, homocysteine.

**Figure 2 nutrients-15-00210-f002:**
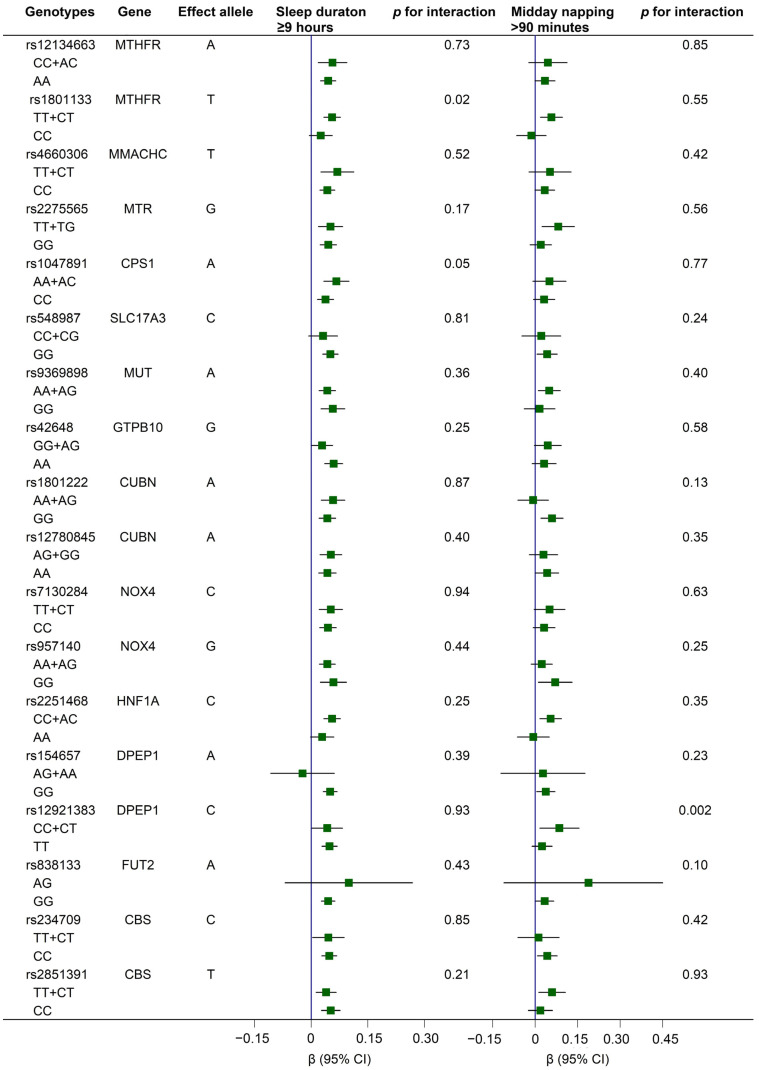
Associations of long sleep duration (≥9 h) and middy napping (>90 min) with serum Hcy levels according to genotypes of 18 Hcy-related variants. Generalized linear regression models were conducted, with adjustments for age (continuous), sex (female, male), education level (primary school or below, middle school, high school or higher), body mass index (continuous), presence of hypertension, dyslipidemia, and diabetes (yes, no), smoking status (current, former, never), drinking status (current, former, never), dietary intake of meat, milk or dairy products, beans or soy products, fish or seafood, and fruits or vegetables (≥5 times/week; yes, no), regular exercise (yes, no), snoring (yes, no), sleep quality (good, fair, poor), sleep duration (continuous), and midday napping (continuous); in addition, sleep duration (continuous) was adjusted for investigation of the gene–napping interaction in Hcy, and midday napping (continuous) was adjusted for investigation of the gene–sleep interaction in Hcy. Interactions were tested by modeling the product terms of genotypes with sleep duration or midday napping (all as continuous variables) into the multivariable-adjusted models, respectively. Note: The rs838133-AA genotype was not detected in the present study. CI, confidence interval; Hcy, homocysteine.

**Table 1 nutrients-15-00210-t001:** Baseline characteristics of the total study population according to sleep duration and midday napping (N = 19,426).

	Sleep Duration, Hours				Midday Napping, Minutes				
	<7	7 to <8	8 to <9	≥9	0	1–30	31–60	61–90	>90
Sample size, *n*	1132	4695	8269	5330	8933	2797	4919	1631	1146
Age, years	63.0 (8.1)	62.4 (7.7)	62.9 (8.1)	63.5 (8.8)	61.9 (8.0)	62.8 (8.1)	64.1 (8.3)	64.5 (8.4)	64.2 (8.5)
Female, (%)	57.9	59.1	59.1	55.5	64.1	62.3	51.5	43.9	48.3
Male, (%)	42.1	40.9	40.9	44.5	35.9	37.7	48.5	56.1	51.7
Education level, (%)									
Primary school or below	17.1	16.2	19.8	26.3	21.5	18.9	19.3	18.1	27.0
Middle school	36.2	36.2	37.8	38.2	38.7	34.1	37.2	36.8	37.6
High school or beyond	46.0	47.0	41.9	35.0	39.2	46.3	43.0	44.8	35.3
BMI, kg/m^2^	24.3 (3.2)	24.1 (3.0)	23.9 (3.07)	23.8 (3.2)	23.9 (3.1)	23.9 (3.1)	24.1 (3.1)	24.0 (3.1)	24.2 (3.1)
eGFR, mL/min/1.73 m^2^	83.8 (16.1)	84.0 (15.4)	83.0 (16.2)	81.7 (17.2)	83.3 (16.3)	83.4 (16.4)	83.0 (15.9)	81.5 (16.9)	82.4 (17.0)
Hypertension, (%)	56.8	58.1	58.3	59.7	54.8	58.4	62.5	62.7	65.5
Dyslipidemia, (%)	41.3	39.7	38.7	38.9	37.0	39.5	40.3	42.0	45.8
Diabetes, (%)	20.1	18.4	19.3	20.4	17.2	20.0	21.2	22.3	23.5
Smoking status, (%)									
Current smoker	19.7	16.0	15.5	17.8	15.6	13.4	17.2	20.3	22.8
Former smoker	10.9	10.0	9.5	11.0	7.7	9.0	12.3	15.2	14.7
Never smoker	69.4	74.1	75.0	71.2	76.7	77.5	70.5	64.5	62.6
Drinking status, (%)									
Current drinker	28.7	26.6	24.9	26.1	23.6	23.4	27.8	30.4	34.8
Former drinker	4.8	4.6	4.4	5.2	3.6	4.7	4.8	8.9	6.4
Never drinker	66.5	68.8	70.7	68.7	72.8	71.9	67.4	60.7	58.8
Dietary intake ^a^, (%)									
Meat	53.7	53.2	53.9	49.8	54.9	52.2	53.0	51.2	52.1
Milk or dairy products	43.4	44.2	43.6	39.5	45.3	40.8	44.4	42.6	39.8
Beans or soy products	56.5	56.4	56.5	52.2	58.3	53.4	57.3	54.8	55.1
Fish or seafood	22.1	20.8	22.9	20.9	23.4	22.3	22.0	19.4	20.6
Fruits or vegetables	96.9	96.9	96.5	96.7	96.9	96.8	96.7	96.0	95.3
Regular exercise ^b^, (%)	71.0	67.8	69.6	63.3	71.8	65.3	70.6	70.8	66.1
Snoring, (%)	44.3	43.1	39.6	38.9	34.4	44.1	45.4	48.3	48.2
Sleep quality, (%)									
Good	30.5	35.2	37.0	38.2	37.5	34.1	35.7	35.1	40.1
Fair	44.3	50.0	50.9	48.8	46.9	51.1	52.7	55.0	47.9
Poor	25.3	14.8	12.1	13.0	15.6	14.8	11.7	9.9	12.0
Job category, (%)									
Manufacturing or manual labor	44.8	44.8	45.6	47.0	43.7	47.9	43.5	43.5	46.7
Service or sales work	28.9	27.7	29.0	28.3	30.6	27.6	29.9	28.3	28.1
Office work	14.0	15.5	12.3	10.3	13.6	11.4	12.9	15.4	10.7
Past shift work, years									
None	45.4	42.2	43.8	42.2	44.2	43.5	44.3	43.9	40.6
≤5.00	13.9	15.7	13.6	14.4	14.9	13.7	14.1	13.9	13.4
5.25–10.00	10.2	9.8	10.9	11.5	10.3	11.3	10.1	10.5	11.6
10.50–20.00	11.7	12.3	11.1	10.9	11.3	11.5	11.2	9.6	11.9
>20.00	7.7	8.9	8.1	7.5	7.9	7.7	7.4	9.7	8.6

BMI = body mass index; eGFR = estimated glomerular filtration rate. Data are presented as mean (standard deviation) for continuous variables and percentages for categorical variables. ^a^ Consumption frequency ≥5 times/week. ^b^ Regular exercise was defined as exercising ≥30 min ≥5 times/week and lasting for at least 6 months.

**Table 2 nutrients-15-00210-t002:** Baseline characteristics of the total participants and participants (N = 19,426) with genetic data (N = 15,126).

	Total Participants	Participants with Genetic Data	*p*
Sample size, *n*	19,426	15126	
Age, years	62.9 (8.2)	63.2 (8.0)	0.006
Female, (%)	11,272 (58.0)	8602 (56.9)	0.03
Male, (%)	8154 (42.0)	6524 (43.1)	0.03
Education level, (%)			0.72
Primary school or below	3998 (20.6)	3143 (20.8)	
Middle school	7272 (37.4)	5724 (37.8)	
High school or beyond	8052 (41.4)	6177 (40.8)	
BMI, kg/m^2^	24.0 (3.1)	24.0 (3.1)	0.96
eGFR, mL/min/1.73 m^2^	82.9 (16.3)	82.8 (16.1)	0.33
Hypertension, (%)	11,375 (58.6)	9018 (59.6)	0.05
Dyslipidemia, (%)	7601 (39.1)	5890 (38.9)	0.73
Diabetes, (%)	3775 (19.4)	2939 (19.4)	0.99
Smoking status, (%)			0.38
Current smoker	3207 (16.5)	2521 (16.7)	
Former smoker	1963 (10.1)	1591 (10.5)	
Never smoker	14,256 (73.4)	11,014 (72.8)	
Alcohol intake status, (%)			0.52
Current drinker	5030 (25.9)	3915 (25.9)	
Former drinker	906 (4.7)	745 (4.9)	
Never drinker	13,490 (69.4)	10,466 (69.2)	
Dietary intake ^a^, (%)			
Meat	10,231 (52.7)	7930 (52.4)	0.84
Milk or dairy products	8245 (42.4)	6347 (42.0)	0.66
Beans or soy products	10,746 (55.3)	8367 (55.3)	0.97
Fish or seafood	4282 (22.0)	3322 (22.0)	0.95
Fruits or vegetables	18,776 (96.7)	14,618 (96.6)	0.99
Regular exercise ^b^, (%)	13,232 (68.1)	10,375 (68.6)	0.63
Snoring, (%)	7877 (40.5)	6220 (41.1)	0.56
Sleep quality, (%)			0.74
Good	7097 (36.5)	5587 (36.9)	
Fair	9651 (49.7)	7473 (49.4)	
Poor	2678 (13.8)	2066 (13.7)	
Job category, (%)			0.36
Manufacturing or manual labor	45.8	45.9	
Service or sales work	28.7	29.1	
Office work	12.4	12.4	
Past shift work, years			0.59
None	43.7	44.2	
≤5.00	14.0	13.8	
5.25–10.00	10.8	11.0	
10.50–20.00	11.3	11.3	
>20.00	7.9	7.9	

BMI = body mass index; eGFR = estimated glomerular filtration rate. Characteristics are presented as mean (standard deviation) for continuous variables and percentages for categorical variables. The *p* values were derived from analysis of variance or Mann–Whitney U tests for continuous variables according to data distribution and χ^2^ tests for category variables. ^a^ Consumption frequency ≥5 times/week. ^b^ Regular exercise was defined as exercise ≥30 min ≥5 times/week and lasting for at least six months.

**Table 3 nutrients-15-00210-t003:** Association of sleep duration and midday napping with serum homocysteine levels.

	Homocysteine ^a^, β (95% Confidence Interval)		
	Model 1 ^b^	Model 2 ^c^	Model 3 ^d^
Sleep duration, hours			
<7	0.004 (−0.024, 0.032)	0.005 (−0.021, 0.031)	0.005 (−0.021, 0.031)
7 to <8	0.000 (ref)	0.000 (ref)	0.000 (ref)
8 to <9	0.024 (0.009, 0.040)	0.013 (−0.001, 0.027)	0.013 (−0.001, 0.027)
≥9	0.079 (0.062, 0.096)	0.046 (0.030, 0.062)	0.045 (0.029, 0.061)
Midday napping, minutes			
0	0.040 (0.021, 0.058)	0.015 (−0.002, 0.032)	0.013 (−0.004, 0.03)
1–30	0.000 (ref)	0.000 (ref)	0.000 (ref)
31–60	0.010 (−0.010, 0.030)	0.005 (−0.014, 0.024)	0.003 (−0.015, 0.022)
61–90	0.006 (−0.020, 0.033)	−0.007 (−0.032, 0.017)	−0.008 (−0.033, 0.016)
>90	0.056 (0.027, 0.086)	0.033 (0.005, 0.060)	0.029 (0.001, 0.057)

^a^ Homocysteine levels were subjected to natural logarithmic transformation to approximate normal distribution. ^b^ Model 1 adjusted for age (continuous) and sex (female, male). ^c^ Model 2 additionally adjusted for education levels (primary school or below, middle school, high school or beyond), body mass index (continuous), estimated glomerular filtration rate (continuous), hypertension (yes, no), dyslipidemia (yes, no), diabetes (yes, no), smoking status (current, ever, never), drinking status (current, ever, never), dietary intake of meat, milk or dairy products, beans or soy products, fish or seafood, and fruits or vegetables (≥5 times/week; yes, no), regular exercise (yes, no), snoring (yes, no), and sleep quality (good, fair, poor). ^d^ Model 3 additionally adjusted for sleep duration or midday napping; each group adjusted for the other covariate except itself.

## Data Availability

Requests may be addressed to the corresponding authors.

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
