# Peer review of "Sleep Duration, Midday Napping, and Serum Homocysteine Levels: A Gene–Environment Interaction Study"

_nutrients, 2023, doi:10.3390/nu15010210_

Round 1

Reviewer 1 Report

This study explores the association of Sleep duration and midday napping with serum homocysteine levels in the Dongfeng-Tongji cohort. They found significant interactions regarding Hcy levels were observed for long sleep duration with GRS and MTHFR rs1801133, and long midday napping with DPEP1 rs12921383. This study concluded that long sleep duration and midday napping were associated with elevated serum Hcy levels, independently and jointly, and amplified the genetic susceptibility to higher Hcy. Overall, this study is well-written and well-conducted. I just some minor suggestions for the considerations of the authors.

1.       Did this study measure shift work and other important covariates?

2.       A subgroup analysis for the key findings with significant interaction in figure 1 may be helpful for the authors to understand the direction and magnitude of associations modulated by genetic factors. From the figure, I have no idea how the genetic factor modulated the association of sleep duration with Hcy level. This may raise concern regarding insufficient sample size.

3.       May the authors like to consider combine sleep duration and midday napping for the 24-hour sleep duration?

Reviewer 2 Report

Congratulations to the authors for the kind of work they have done.
It is very articulated and full of information. The tables are very clear and I would suggest inserting the descriptive tables of the sample not in the Supplementary but at the beginning of the results. Statistical tests are appropriate.
ADDITIONAL MATERIAL Why in supplementary tables 1 and 2 are there only values ​​for women and no men?
